# Long-term effects of bariatric surgery on acute kidney injury: a propensity-matched cohort in the UK Clinical Practice Research Datalink

Uwe Koppe,[1,2] Dorothea Nitsch,[1,3] Kathryn E Mansfield,[1] Rohini Mathur,[1] Krishnan Bhaskaran,[1] Rachel L Batterham,[4,5,6] Liam Smeeth,[1] Ian J Douglas[1]

[1]Faculty of Epidemiology and Population Health, London School of Hygiene and Tropical Medicine, London, UK
[2]Department of Infectious Disease Epidemiology, Robert Koch Institute, Berlin, Germany
[3]Royal Free London NHS Trust, London, UK
[4]Centre for Obesity Research, Rayne Institute, Department of Medicine, University College London, London, UK
[5]University College London Hospital Bariatric Centre for Weight Management and Metabolic Surgery, London, UK
[6]National Institute of Health Research, University College London Hospital Biomedical Research Centre, London, UK

**Correspondence to**
Dr Uwe Koppe; koppeu@rki.de

## ABSTRACT

**Objective** Bariatric surgery is an effective method of weight reduction and has been associated with acute kidney injury (AKI) as a perioperative event. However, the long-term effects of the weight reduction after surgery on AKI are unknown. The objective of this study is to quantify the association of bariatric surgery with later risk of AKI.

**Design** This study uses a propensity score-matched cohort of patients from the UK Clinical Practice Research Datalink database with and without bariatric surgery to compare rates of AKI episodes derived from linkage to the Hospital Episode Statistics.

**Setting** England, UK.

**Participants** We included 2643 patients with bariatric surgery and 2595 patients without.

**Results** Results were compatible with an increased risk of AKI in the first 30 days following surgery compared with patients without surgery, but AKI incidence was substantially decreased in patients with bariatric surgery during long-term follow-up (rate ratio 0.37, 95% CI 0.23 to 0.61) even after accounting for chronic kidney disease status at baseline. Over the whole period of follow-up, bariatric surgery had a net protective effect on risk of AKI (rate ratio 0.45, 95% CI 0.28 to 0.72).

**Conclusions** Bariatric surgery was associated with protective effects on AKI incidence during long-term follow-up. While the risk of AKI may be increased within the first 30 days, the net effect seen was beneficial.

## INTRODUCTION

The proportions of overweight and obese adults in England in 2014 are estimated to be 61.7% and 25.6%, respectively, and are increasing over time.[1] Obesity is associated with serious health consequences including type 2 diabetes mellitus (T2DM), cardiovascular diseases, cancers and chronic kidney disease (CKD).[2–4] Bariatric surgery has been shown to be a highly effective intervention for achieving weight loss and reducing the burden of comorbidities, such as T2DM, metabolic syndrome and hypertension.[5 6] A recent observational study on recipients of bariatric surgery from the UK confirmed sustained

### Strengths and limitations of this study

► This study uses high-quality data from linked databases in England (Clinical Practice Research Datalink and Hospital Episode Statistics) to describe long-term effects of bariatric surgery on acute kidney injury (AKI) for the first time.

► Data are captured prospectively and continuously, thus allowing follow-up of patients over long-time periods.

► Outcome measures are obtained with standardised International Classification of Diseases, 10th Revision codes, which have been shown to accurately identify AKI, but do not allow grading of severity and may therefore underestimate true AKI incidence.

► Only AKI events recorded during a hospital admission were included in the analysis likely representing the more serious events of AKI.

► The study population was mostly female, of middle age and had a history of type 2diabetes mellitus. Thus, the results might not be applicable for other groups suffering from obesity such as adolescents.

weight loss as well as resolution of T2DM and hypertension over a period of 4years.[7]

Acute kidney injury (AKI) is defined as a sudden (over hours or days) drop in kidney function characterised by increased serum creatinine and/or reduced urine output. AKI has been linked to increased in-hospital mortality, length of hospital stay and subsequent development of CKD.[8] While T2DM, CKD and obesity have been described as risk factors for AKI, it can also be precipitated by nephrotoxic drugs, surgical interventions and sepsis.[8–10] AKI has been described as a short-term complication of bariatric surgery, stemming from rhabdomyolysis.[10–16] In addition, AKI has been linked to nephrolithiasis, which can develop over time after Roux-en-Y gastric bypass surgery.[11 17] To the best of our knowledge, no studies have been published

examining the long-term effects of bariatric surgery on AKI.

In this study, we investigate the long-term effects of bariatric surgery on AKI to see whether the expected reduction in body mass index (BMI) has any impact on subsequent renal health. We used routinely collected electronic health record data from primary and secondary care. For this, we conducted a matched cohort study using prospectively collected data from patients in the UK Clinical Practice Research Datalink (CPRD).

## METHODS

### Patient and public involvement

Patients or public were not involved in the design or conduct of the study.

### Study design

We undertook a matched cohort study using prospectively collected data from CPRD patients registered before 31 December 2014 linked to the Hospital Episodes Statistics (HES) database to investigate long-term effects of bariatric surgery on AKI.

### Data source

The CPRD database contains anonymised, routinely collected data on approximately 10 million patients in participating primary care practices in the UK, including demographic characteristics, current and previous diagnoses, prescribing, test results and lifestyle factors. Diagnoses, signs and symptoms are recorded using Read codes.[18] Patients are broadly representative of the UK population and the data have been validated for a wide range of outcomes.[19–21] The HES database contains patient data from hospital admissions to English hospitals within the National Health Service.[22] For each hospital admission, the diagnoses are recorded using standardised codes of the International Classification of Diseases, 10th Revision (ICD-10).[23 24] Data from 70% of CPRD practices in England have been linked at patient level with HES admission data, thus allowing the combined analysis of data from primary and acute hospital care for a subset of patients.[19]

### Cohort design and propensity matching

A detailed description of how the cohort was constructed is described elsewhere.[7] In brief, records of patients who underwent bariatric surgery (n=3882) between 1997 and 2015 were matched to individuals who did not undergo surgery (n=3882) using propensity scores.

Study population matching and the propensity score incorporated information on age, sex, calendar period, history of T2DM, hypertension, coronary heart disease, cerebrovascular disease, peripheral vascular disease, other atheroma, use of insulin, use of oral antidiabetic medication, use of statins, smoking status and alcohol consumption.

Patients with bariatric surgery were identified using Read codes for surgery in the CPRD database (online supplementary appendix S1) and were included in the study if they had been registered in the CPRD ≥12 months prior to the intervention. We excluded those with a record of prior bariatric surgery reversal.

For the comparison group, the inclusion criteria were to have at least one BMI measurement ≥40 kg/m$^2$ during their CPRD registration, which could span 10 years or more, ≥12 months of follow-up prior to the index date in the database, and no prior record of bariatric surgery or bariatric surgery reversal. Based on this, it is therefore possible that the BMI recorded closest to the index date was lower than 40 kg/m².

The study sample was restricted to eligible patients registered at practices linked to the HES database and information on AKI events was obtained, resulting in a final cohort comprising 2643 patients who underwent bariatric surgery, and 2595 patients who did not.

Follow-up started on the day of surgery for those with bariatric surgery, and for the comparison group who did not undergo bariatric surgery, on the surgery date of their matched case. Patient records were censored at the earliest of: AKI, death, leaving the practice, latest data collection from current practice or end of linkage period to the HES database.

### Outcomes and covariates

The primary outcome of this study was the incidence rate of the first AKI episode during follow-up in patients with and without bariatric surgery. AKI episodes were obtained from the HES database using ICD-10 codes: N17.0 ('acute kidney failure with tubular necrosis'), N17.1 ('acute renal failure with acute cortical necrosis), N17.2 ('acute renal failure with medullary necrosis'), N17.8 ('other acute renal failure'), N17.9 ('acute kidney failure, unspecified') and N19 ('unspecified kidney failure'). In this cohort, events coded with N17.1, N17.2 and N17.8 were not found. AKI events that occurred before the start of follow-up were recorded as a binary variable 'history of AKI', while AKI events occurring during follow-up were used to analyse AKI incidence.

Recorded serum creatinine values from the CPRD database were not routinely standardised with isotope dilution mass spectrometry before 2013. Thus, we assumed all measurements to be unstandardised and multiplied the creatinine measures with the factor 0.95 before calculating the estimated glomerular filtration rate (eGFR) using the 'Chronic Kidney Disease Epidemiology Collaboration' equation.[25] Ethnicity was not considered in the eGFR calculation due to incomplete recording in the database and the low proportion of Afro-Caribbean people in the population. CKD stages were defined according to eGFR values in mL/min/1.73 m$^2$ according to current guidelines[26]: eGFR ≥60=no known CKD; eGFR 45–59=stage 3a; eGFR 30–44=stage 3b; eGFR 15–29=stage 4; eGFR <15=stage 5. Baseline CKD status was derived from eGFR measurements in the year prior to start of

follow-up by: (1) taking the last two measurements before the index date ≥90 days apart—with the higher eGFR value corresponding to the CKD baseline status or (2) taking the most recent serum creatinine result if only one suitable test result was available. Since serum creatinine is more likely to be tested in the acutely unwell or in people who are routinely monitored as part of incentivised programmes (eg, people with diabetes), patients without measurements of CKD baseline status were assumed to have no CKD[27] and were analysed as such.

## Statistical analysis

Though propensity score matching was employed to minimise confounding, we compared the distribution of baseline characteristics between the exposed and unexposed groups to check for any imbalances that may be relevant to the outcome of AKI. The baseline distribution of categorical variables was analysed using percentages and $\chi^2$ tests. Continuous variables were analysed as means with SD for normally distributed variables and medians with IQR for non-normally distributed variables. Differences in continuous variables were analysed with Student's t-tests or Wilcoxon rank-sum tests for normally and non-normally distributed data, respectively.

The association between bariatric surgery and AKI was analysed using a Poisson regression model with a time to first event analysis. P values were calculated using Wald tests. In order to separate short-term effects of the surgery from potential long-term effects, we analysed the association separately for: (1) events within the first 30 days and (2) events after 30 days. When the cohort was initially constructed, propensity score matching was used to deal with confounding.[7] This study uses a subset of this cohort since patients from practices without linkage between the CPRD and HES databases had to be excluded (as AKI was assessed using hospital admission data). To identify variables for the multivariable model, potential confounders that were not deemed to be on the causal pathway were added individually to the univariable model. If the addition changed the effect estimate ≥10% these variables were included in the multivariable model. Consequently, history of AKI, history of taking oral antidiabetics and BMI at baseline were included (online supplementary appendix S2). In addition, age at baseline, sex, calendar period (1997–2005, 2006–2010, 2011–2015) and CKD status at baseline were selected a priori as forced variables. For models with <40 outcomes, only age and sex were included in the multivariable model due to data sparsity.

The 5% bands of patients with the highest and lowest propensity scores were excluded from the primary analysis ('trimming') since these contain patients who are treated in stark contrast to their health status, potentially causing bias.[28]

Heterogeneity of effect estimates between the calendar periods was tested with a Likelihood Ratio Test.

The analysis was performed for all patients with bariatric surgery and also further stratified by type of surgery.

Patients with stage 5 CKD (baseline eGFR <15 mL/min/1.73 m$^2$) were excluded from the analyses since this constitutes end-stage renal disease. In addition, patients with missing data in ≥1 variable of the multivariable model were excluded from both univariable and multivariable analyses.

All analyses were performed with Stata V.14.1.

## Subgroup analyses

Several planned sensitivity analyses were undertaken: (1) To determine the net effect of the intervention we calculated the risk of AKI over the whole period of follow-up; (2) The prevalence of decreased kidney function in the CPRD database was similar to that in a nationally representative kidney disease registry[27] indicating that patients with missing eGFR measurements are unlikely to have CKD. To identify potential differences in the effect between patients with known and unknown eGFR measurements, we restricted the analysis to (a) patients known to have no CKD at baseline (baseline eGFR ≥60 mL/min/1.73 m$^2$), (b) patients without known CKD at baseline (as above but including patients with missing creatinine values at baseline and assuming these individuals to have no CKD) and (c) patients with known CKD at baseline.[27] (3) Moreover, to investigate the effect in a group of particular interest which is under more scrutiny for measuring kidney function we restricted the analysis to patients with: (a) T2DM and (b) a history of taking insulin; (4) To avoid misclassification of low eGFR values as AKI,[29] we excluded patients with stage 4 CKD at baseline; (5) We restricted the analysis to ICD-10 codes N17.0 and N17.9, which have a high positive predictive value for AKI[24]; (6) We increased the immediate postsurgery time span from 30 to 60 days; (7) We included people with extreme propensity scores and (8) We excluded patients with a BMI <35 kg/m² at baseline.

## Results

Since linkage to the HES database was only possible for patients whose general practitioners (GPs) had agreed for their practice data to be linked to HES (online supplementary appendix S3), there were 2643 patients with bariatric surgery and 2595 people without surgery resulting in a cohort of overall 5238 people with a median follow-up of 2.9 years (table 1). The median follow-up prior to baseline was similar between the groups: 8.8 years (IQR: 8.1 years) for patients with bariatric surgery and 9.3 years (IQR: 8.0 years) for people without surgery.

This cohort was comparable with the cohort from the original study regarding sex, mean age, mean BMI, history of T2DM, type of bariatric surgery and the imbalance of BMI at baseline.[7] More patients in the intervention group had a history of AKI compared with the comparison group (1.1% vs 0.4%). Of the 106 included events during follow-up, 84.9% were classified with the ICD-10 code N17.9 ('acute kidney failure, unspecified'), 12.3% were coded as N19 ('unspecified kidney failure') and 2.8% had a code of N17.0 ('acute kidney failure with

**Table 1** Baseline data for CPRD/HES-linked cohort study of people with bariatric surgery and the corresponding propensity score-matched* comparison cohort (data are n (%) unless otherwise specified)

| | Bariatric surgery (n=2643) | Matched comparison group without surgery (n=2595) | P values† |
|---|---|---|---|
| Follow-up (years), median (IQR) | 2.9 (3.2) | 2.9 (3.4) | 0.616 |
| Age (years), mean (SD) | 45.2 (10.7) | 45.0 (10.8) | 0.417 |
| 17–39, n (%) | 818 (31.0) | 826 (31.8) | 0.727 |
| 40–49, n (%) | 945 (35.8) | 928 (35.8) | |
| 50–85, n (%) | 880 (33.3) | 841 (32.4) | |
| BMI at baseline, mean (SD) | 44.9 (8.9) | 42.2 (6.5) | <0.001 |
| 13–34, n (%) | 297 (11.2) | 287 (11.1) | <0.001 |
| 35–39, n (%) | 448 (17.0) | 456 (17.6) | |
| 40–44, n (%) | 625 (23.7) | 1118 (43.1) | |
| 45–49, n (%) | 571 (21.6) | 438 (16.9) | |
| 50–94, n (%) | 667 (25.2) | 253 (9.8) | |
| Missing, n (%) | 35 (1.3) | 43 (1.7) | |
| Female | 2131 (80.6) | 2131 (82.1) | 0.166 |
| History of | | | |
| Cerebrovascular disease | 37 (1.4) | 26 (1.0) | 0.186 |
| Coronary heart disease | 104 (3.9) | 82 (3.2) | 0.130 |
| Peripheral vascular disease | 11 (0.4) | 15 (0.6) | 0.405 |
| Other atheroma | 0 | <5‡ | 0.313 |
| T2DM | 900 (34.1) | 853 (32.9) | 0.365 |
| Taking oral antidiabetic | 571 (21.6) | 455 (17.5) | <0.001 |
| Taking insulin | 180 (6.8) | 156 (6.0) | 0.238 |
| Hypertension | 890 (33.7) | 869 (33.5) | 0.886 |
| Statin use | 699 (26.4) | 640 (24.7) | 0.139 |
| AKI | 30 (1.1) | 11 (0.4) | 0.003 |
| Alcohol status | | | |
| Non-drinker | 435 (16.5) | 397 (15.3) | 0.366 |
| Ex-drinker | 278 (10.5) | 236 (9.1) | |
| Current drinker (amount unknown) | 15 (0.6) | 13 (0.5) | |
| <2 units/day | 659 (24.9) | 644 (24.8) | |
| 3–6 units/day | 862 (32.6) | 909 (35.0) | |
| >6 units/day | 170 (6.4) | 164 (6.3) | |
| Unknown | 224 (8.5) | 232 (8.9) | |
| Smoking status | | | |
| Non-smoker | 1126 (42.6) | 1151 (44.4) | 0.093 |
| Current smoker | 403 (15.3) | 345 (13.3) | |
| Ex-smoker | 1112 (42.1) | 1099 (42.4) | |
| Unknown | <5‡ | 0 | |
| CKD at baseline | | | |
| Baseline CKD status absent | 1119 (42.3) | 1299 (50.1) | <0.001 |
| No CKD | 1470 (55.6) | 1242 (47.9) | |
| Stage 3a | 27 (1.0) | 37 (1.4) | |
| Stage 3b | 16 (0.6) | 10 (0.4) | |
| Stage 4 | 10 (0.4) | 5 (0.2) | |
| Stage 5 | <5‡ | <5‡ | |

**Table 1** Continued

| | Bariatric surgery (n=2643) | Matched comparison group without surgery (n=2595) | P values† |
|---|---|---|---|
| Type of bariatric surgery | | | |
| Gastric band | 1193 (45.1) | | |
| Sleeve gastrectomy | 364 (13.8) | | |
| Gastric bypass | 1075 (40.7) | | |
| Other | 11 (0.4) | | |
| ICD-10 code for AKI during follow-up | n=44 | n=62 | |
| N17.0 (acute kidney failure with tubular necrosis) | <5‡ | <5‡ | 0.927 |
| N17.9 (acute kidney failure, unspecified) | 38 (86.4) | 52 (83.9) | |
| N19 (unspecified kidney failure) | 5 (11.4) | 8 (12.9) | |

*In the original study, each surgery patient was matched 1:1 to the person without surgery with the closest propensity score, choosing matches at random where more than one possible match had the same score.[7]

†Categorical variables: $\chi^2$ test; continuous variables: Student's t-test+SD if normally distributed, Wilcoxon rank-sum test+IQR if non-normally distributed.

‡Cell counts <5 have been suppressed to ensure anonymity.

AKI, acute kidney injury; BMI, body mass index; CKD, chronic kidney disease; CPRD, Clinical Practice Research Datalink; HES, Hospital Episodes Statistics; ICD-10, International Classification of Diseases, 10th Revision; T2DM, type 2 diabetes mellitus.

tubular necrosis'). CKD status at baseline was unknown for about half of the patients in each group with a slightly higher proportion in the unexposed group (50.1% vs 42.3%). The majority of the patients with creatinine tests at baseline did not have CKD (96.2%).

The number of AKI events recorded in the first 30 days of follow-up was low. All five events happened in patients with bariatric surgery and none were recorded in the control group, which is consistent with the possibility of an increased risk of AKI directly after surgery (table 2).

From 30 days onwards, bariatric surgery had a protective association with AKI risk (crude rate ratio (RR) 0.62, 95% CI 0.40 to 0.95). The effect estimate of the multivariable model indicated an even stronger protective effect associated with bariatric surgery (RR 0.37, 95% CI 0.23 to 0.61), largely due to the confounding by AKI prior to baseline.

The analysis by type of surgery yielded protective effect estimates for all types but the CIs were wide and no comparison between individual procedures was feasible. Sensitivity analyses yielded similar results (online supplementary appendix S4). A sensitivity analysis restricted to patients with known CKD at baseline could not be done owing to sparse data. Investigation of the effect of bariatric surgery over the whole follow-up period resulted in a protective net effect associated with the intervention in univariable (RR 0.71, 95% CI 0.47 to 1.07) and multivariable (RR 0.45, 95% CI 0.28 to 0.72) analyses.

## DISCUSSION

In this study using prospectively recorded routine healthcare data from a representative sample in the UK, bariatric surgery was associated with a potentially increased risk of AKI within the first 30 days after surgery (five events in patients with bariatric surgery, no events in control patients) but a strongly protective association thereafter (adjusted RR 0.37, 95% CI 0.23 to 0.61). The association was consistent across subgroups and sensitivity analyses. To the best of our knowledge, this is the first study to describe long-term effects of bariatric surgery on AKI.

AKI has been described as a perioperative event for bariatric surgery.[12 13 15 16] Our results are consistent with an increased risk in the early stages after surgery, however, our analysis lacked enough early events to rule out chance as a reason for the results observed. Since patients do not have kidney function measures routinely checked by their family physician after bariatric surgery, many events could remain unnoticed. Patients with known CKD are more thoroughly checked for AKI and are a valuable subgroup to investigate, but the numbers in this dataset were too low to analyse.

This study uses high-quality data from routine medical care in the UK. The healthcare system allows universal patient access to primary and secondary care so that the data are representative of the population. Patients are followed continuously while they are registered with a GP allowing prospective data capture over long observation periods and avoiding problems with reverse causality.

Some limitations need to be considered. Even though the data are taken from a representative sample of the UK population, the baseline data indicate that patients who undergo bariatric surgery are mostly female, of middle age and with a history of T2DM. While the results were adjusted for age and sex, they might not be applicable

**Table 2** Association of bariatric surgery with first incident AKI, stratified by length of follow-up

| | PY | Events | Rate per 1000 PY (95% CI) | Crude RR (95% CI)* | P values† | Adjusted RR (95% CI)‡ | P values† |
|---|---|---|---|---|---|---|---|
| **All patients** | | | | | | | |
| Day 1–30 | | | | | | | |
| Unexposed | 203 | 0 | 0 | – | | | |
| Bariatric surgery | 199 | 5 | 25.1 (10.5 to 60.4) | – | | | |
| >Day 30 | | | | | | | |
| Unexposed | 7882 | 54 | 6.9 (5.2 to 8.9) | – | | | |
| Bariatric surgery | 8061 | 34 | 4.2 (3.0 to 5.9) | 0.62 (0.40 to 0.95) | 0.027 | 0.37 (0.23 to 0.61) | <0.001 |
| **All patients analysed by type of surgery§** | | | | | | | |
| Day 1–30 | | | | | | | |
| Unexposed | | | | | | | |
| Gastric band | | | | | | | |
| Sleeve gastrectomy | | | | | | | |
| Gastric bypass | | | | | | | |
| Other | | | | | | | |
| >Day 30 | | | | | | | |
| Unexposed | 7882 | 54 | 6.9 (5.2 to 8.9) | – | | | |
| Gastric band | 4614 | 17 | 3.7 (2.3 to 5.9) | 0.54 (0.31 to 0.93) | 0.026 | | |
| Sleeve gastrectomy | 728 | <5¶ | 5.5 (2.1 to 14.6) | 0.80 (0.29 to 2.21) | 0.670 | | |
| Gastric bypass | 2655 | 13 | 4.9 (2.8 to 8.4) | 0.71 (0.39 to 1.31) | 0.277 | | |
| Other | 63 | 0 | – | – | | | |
| **All patients over whole period of follow-up** | | | | | | | |
| Unexposed | 8085 | 54 | 6.7 (5.1 to 8.7) | – | | | |
| Bariatric surgery | 8259 | 39 | 4.7 (3.5 to 6.5) | 0.71 (0.47 to 1.07) | 0.099 | 0.45 (0.28 to 0.72) | 0.001 |

Unexposed refers to the propensity-matched comparison group.
*Poisson regression model.
†Wald test.
‡Poisson regression model adjusted for age at baseline, sex, calendar time, CKD at baseline, history of AKI, history of taking oral antidiabetics and BMI at baseline.
§No analysis for day 1–30 owing to sparse data.
¶Cell counts <5 have been suppressed to ensure anonymity.
AKI, acute kidney injury; BMI, body mass index; CKD, chronic kidney disease; PY, person years; RR, rate ratio.

for other groups suffering from obesity like adolescents. Linkage between the CPRD and HES databases was restricted to England. However, there is no cogent reason why the results should not be applicable to regions with similar healthcare systems, both in the UK and internationally. We had insufficient data to determine whether the association with AKI varied between different types of bariatric surgery; we found a protective effect for gastric band, but results were inconclusive for sleeve gastrectomy and gastric bypass.

Of all AKI episodes identified in the HES database, the ICD-10 codes N17.0 and N17.9 comprised 87.7% of all events and have previously been shown to accurately identify AKI in a single-centre study.[24] However, there are no ICD-10 codes for grades of AKI severity. Thus, we were not able to investigate whether there was an association between bariatric surgery and AKI severity. In general, AKI diagnosed during hospitalisation is likely to represent more serious AKI events and therefore may underestimate AKI incidence. Thus, the conclusions drawn from this study may only be applicable to severe AKI diagnosed during hospitalisation, however, we would argue these are the most clinically relevant outcomes. Moreover, a patient who experienced a previous AKI episode might be under more scrutiny for detection of future episodes. Since more patients in the bariatric surgery group had a history

of AKI they might have a higher chance of detection of an AKI episode during follow-up, which we adjusted for in our analyses.

Misclassification of diagnostic codes is likely non-differential between the bariatric surgery patients and the matched comparison group and would bias the effect towards the null value. However, it is also conceivable during the immediate postoperative period those undergoing bariatric surgery might have been under more scrutiny to detect potential AKI events than people without surgery. In this case, our current relative risk estimates for the immediate postoperative period would be an overestimate. Another problem of primary care data is that not every patient is routinely checked for their kidney function, as incentives of testing apply primarily for those at risk of kidney disease due to diabetes and hypertension. The study relied on AKI events recorded in HES as part of a hospital admission and over time, the awareness of the importance of AKI has likely changed resulting in secular changes in recording of AKI[23]; analyses have adjusted for calendar period to account for this.[23 30] Future studies with hospital creatinine data should compare the AKI severity between the groups to investigate this issue.

In addition, CKD status at baseline was missing in almost half of the patient population. However, a recent study indicated that the prevalence of CKD in the CPRD database was comparable to that found in nationally representative registry studies.[27] This indicates that patients without a GP record of eGFR measurements at baseline are unlikely to have CKD. In addition, sensitivity analyses investigating the effect in patients with known or unknown CKD status at baseline yielded comparable results.

Since access to bariatric surgery is restricted within the UK healthcare system, some patients might have funded their operation privately, resulting in selection bias. In a recent analysis about 40% of bariatric surgery operations in the UK were privately funded.[31] Thus, the intervention group might have a higher socioeconomic status than the non-exposed group, in which similar patients would not be able to afford surgery. Since the socioeconomic background is an important determinant of health outcomes and was an unmeasured potential confounder not considered in the matching process, this could have led to more positive health outcomes in the intervention group irrespective of surgery and to an overestimation of the effect. In this study setting, it was not possible to determine which patients had privately funded surgery.

Even though most baseline variables were evenly distributed due to the matching process, this does not guarantee that unmeasured variables are evenly distributed as well, which can constitute residual confounding. Incorrect, imprecise or missing measurements of covariates could also have led to residual confounding. For the multivariable model, adjusting for history of AKI led to the strongest change of the effect estimate. AKI events are likely under-recorded in the HES database, for reasons described above, and thus residual confounding is possible. Since adjusting for AKI history led to a stronger effect estimate, the protective effect we report here may be an underestimate if AKI history is missing to the same degree in surgery and non-surgery patients.

This study adds to the evidence of long-term effects of bariatric surgery, and appears to be the first study to quantify a long-term beneficial effect on AKI. Future studies with higher patient numbers may be able to determine differences in effect between types of surgery, investigate the effect in patients with CKD and elucidate mechanisms of the association between bariatric surgery and AKI.

**Contributors** UK, DN, RLB, IJD and LS were responsible for conceptualisation of the study and formulate the research goals and aims. UK, DN, KEM, RM, KB, RLB, LS and IJD developed the methodology and models. UK, KEM, KB, IJD and RM worked on the data curation. UK performed the statistical analysis and wrote the original draft. UK, DN, KEM, RM, KB, RLB, LS and IJD reviewed and commented the draft and gave input on editing.

**Funding** RM is supported by a Sir Henry Wellcome Postdoctoral Fellowship from the Wellcome Trust (201375/Z/16/Z). KB holds a Sir Henry Dale fellowship jointly funded by the Wellcome Trust and the Royal Society (107731/Z/15/Z). RLB is an NIHR Research Professor and supported by funding from the Rosetrees Trust and the Sir Jules Thorn Charitable Trust. LS is supported by a senior clinical fellowship from the Wellcome Trust (098504/Z/12/Z). IJD is funded by an unrestricted grant from GlaxoSmithKline.

**Disclaimer** None of the funders had any involvement in the design of the study, the data collection and analysis, the writing of the report or the decision to submit the paper for publication.

**Competing interests** None declared.

**Patient consent** Not required.

**Ethics approval** This study was approved by the London School of Hygiene & Tropical Medicine Ethics Committee (LSHTM MSc Ethics Ref: 11065) and the Independent Scientific Advisory Committee on Medicines and Healthcare Products Regulatory Agency database research (approval number: 16_106R).

**Provenance and peer review** Not commissioned; externally peer reviewed.

**Data sharing statement** The data were obtained from the Clinical Practice Research Datalink (CPRD). CPRD data governance does not allow us to distribute patient data to other parties. Researchers may apply for data access at www.CPRD.com. The codes used to produce the data for this study are provided in the supporting information.

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
