## [Reviewer comments · BMJ Open]

ARTICLE DETAILS

TITLE (PROVISIONAL)	Long-term effects of bariatric surgery on acute kidney injury: A propensity-matched cohort in the United Kingdom Clinical Practice Research Datalink
AUTHORS	Koppe, Uwe; Nitsch, Dorothea; Mansfield, Kathryn; Mathur, Rohini; Bhaskaran, Krishnan; Batterham, Rachel; Smeeth, Liam; Douglas, Ian

VERSION 1 – REVIEW

REVIEWER	Prof. Dr. Helmut Schiffel Department of Internal Medicine IV(Nephrology) , Uni-versity Hopsital Munich, Munich, Germany
REVIEW RETURNED	07-Nov-2017

GENERAL COMMENTS	There are number of seriuos limitations causing doubts on the reliability of the data. ICD -Codes of AKI are subject to all souirces of bias, . underestimate the real AKI incidence and allow no garding of AKI severity. To examine the effects of bariatric surgery and reduction of body weight on AKI incidence the use of serum creatinine criteria and urine out is mandatory. Amazingly, significant number of patients underwent surgery without pre-operative dtermination of renal function. Surprisingly, the non-intervention group had more AKI hospitalisations than the treated group. The authors should give an convincing explanation . Taken together, the conclusions of the authors are not in accordance with the data. With only 5 documented events after surgery(" the risk of AKI may be increased ") of 2.643 patients there is no clinically relevant risk. The authors statement on the pathophysiology of AKI after bariatric surgery (rhabdomyolysis, nephrolithiasis) is obsolet. The authors should contact a nephrologist
--

REVIEWER	Alex Chang Geisinger Health System USA
REVIEW RETURNED	29-Nov-2017

GENERAL COMMENTS	The authors examine the association between bariatric surgery and risk of AKI using a propensity-score matched cohort of patients from the UK. While this is the best attempt to date at answering this question, I have a few concerns about the matching process that need to be addressed. 1 - Why was BMI not included in the propensity score? BMI clearly is an important factor for considering bariatric surgery. Also, I wonder about the requirement of BMI >= 40 kg/m2 for the comparison group when some patients with BMI >= 35 kg/m2 may have surgery. This
--

	is even more confusing to me when I look at Table 1 and see that 11% of bariatric surgery and non-surgery group patients had BMI between 13-34? And why do some patients have missing baseline BMI? 2 - The authors do point out the limitations of AKI ascertainment with ICD codes, and how it is dependent on testing. Baseline AKI differed between the groups and is also dependent on the length of follow-up prior to the baseline period. Did this differ between groups? One could consider using time of follow-up before the baseline period as a matching variable or a metric capturing exposure to the health system (e.g. # of outpatient visits, hospitalizations) 3 - As the authors point out a major limitation was the lack of baseline CKD status in many patients and also the fact that 40% of patients get bariatric surgery in the private setting. Can the authors comment on the chances these individuals are receiving follow-up care in the private setting as well. Is there any surrogate for socioeconomic status in the dataset that can be assessed? For limitations, misclassification of diagnostic codes could be differential if there is a pre-conceived notion that bariatric surgery patients are at increased risk of AKI. Lastly, why is baseline eGFR missing for patients undergoing bariatric surgery? Is it possible they may have labs drawn elsewhere outside of the NHS? If so, then might they get follow-up labs also outside of the NHS?
--	---

VERSION 1 – AUTHOR RESPONSE

Reply to author's comments

Reviewer: 1

Reviewer Name: Prof. Dr. Helmut Schiffl

Institution and Country: Department of Internal Medicine IV(Nephrology) , University Hospital Munich, Munich, Germany

Please state any competing interests or state 'None declared': None declared

Please leave your comments for the authors below

There are number of serious limitations causing doubts on the reliability of the data. ICD -Codes of AKI are subject to all sources of bias, underestimate the real AKI incidence and allow no grading of AKI severity. To examine the effects of bariatric surgery and reduction of body weight on AKI incidence the use of serum creatinine criteria and urine out is mandatory.

Reply: We agree that analyses based on ICD-10 codes rather than measures of serum creatinine and urine output will likely identify the more severe cases and might underestimate the absolute risk for acute kidney injury (AKI). However, a previous study showed that ICD-10 codes accurately identify AKI with a high positive predictive value of 95% (Tomlinson et al., BMC Nephrol 2013;14:58). Moreover, while grading of AKI severity is not possible, we would argue that AKI identified via ICD-10 codes is clinically relevant since it identifies patients who had to be hospitalised for AKI. Importantly, the relative risk estimates will remain unbiased since the underestimation of absolute risk applies both to patients with and without bariatric surgery. Thus our analysis does yield a valid estimate of the relative risk of AKI requiring hospitalisation in patients with bariatric surgery compared to patients without surgery.

Amazingly, significant number of patients underwent surgery without pre-operative determination of renal function.

Reply: In our analyses, 42.3% of the patients with bariatric surgery did not have serum creatinine values recorded in their primary care record at baseline and we were therefore unable to derive information on any underlying chronic kidney disease (CKD). However, these patients are likely to have had preoperative assessments of kidney function in hospital but this would generally not have been registered in the general practitioner's (GP) record and thus the CPRD database. In addition, a recent study showed that GP's in the UK are good at identifying patients with reduced kidney function (Iwagami et al., *Nephrol Dial Transplant* 2017; 32 (Suppl 2): ii142-ii150) indicating that patients without tests of kidney function in CPRD records are unlikely to have CKD stages 3-5.

Surprisingly, the non-intervention group had more AKI hospitalisations than the treated group.

Reply: We are not sure why this was surprising; determining the relative incidence of AKI hospitalisations among patients with and without bariatric surgery was our primary research question. Whilst the answer was uncertain at the outset, our hypothesis was that people who remain very obese will be more likely to be susceptible to AKI when compared to those who underwent bariatric surgery, as both obesity and CKD are considered risk factors for AKI.

The authors should give a convincing explanation. Taken together, the conclusions of the authors are not in accordance with the data. With only 5 documented events after surgery (" the risk of AKI may be increased") of 2.643 patients there is no clinically relevant risk.

Reply: We respectfully assume that this comment is based on a misunderstanding. We counted five AKI events within the first 30 days of follow-up. However, during all follow-up we counted 106 events overall: 44 in patients with bariatric surgery and 62 in patients without bariatric surgery. The five events within the first 30 days of follow-up were observed only in patients with bariatric surgery. These events represent significant AKI events.

The authors' statement on the pathophysiology of AKI after bariatric surgery (rhabdomyolysis, nephrolithiasis) is obsolete. The authors should contact a nephrologist!

Reply: We would like to politely mention that our co-author Prof. Dorothea Nitsch is an experienced practising nephrologist. She is involved in epidemiological research on nephrology and also holds a clinical contract as an Honorary Consultant Nephrologist with the Royal Free London NHS Foundation trust. We cited previous literature on this topic but agree that with appropriate perioperative management these complications are now less common.

Reviewer: 2

Reviewer Name: Alex Chang

Institution and Country: Geisinger Health System, USA

Please state any competing interests or state 'None declared': none declared

Please leave your comments for the authors below

The authors examine the association between bariatric surgery and risk of AKI using a propensity-score matched cohort of patients from the UK. While this is the best attempt to date at answering this question, I have a few concerns about the matching process that need to be addressed.

1 - Why was BMI not included in the propensity score? BMI clearly is an important factor for considering bariatric surgery. Also, I wonder about the requirement of BMI ≥ 40 kg/m² for the comparison group when some patients with BMI ≥ 35 kg/m² may have surgery. This is even more confusing to me when I look at Table 1 and see that 11% of bariatric surgery and non-surgery group patients had BMI between 13-34? And why do some patients have missing baseline BMI?

Reply: Although BMI is not included in the propensity score, it is adjusted for in the multivariable model and so we are confident that differences between the groups in terms of AKI relative incidence are not driven by differences in BMI.

The larger pool of eligible non-surgery patients was identified based on a recorded BMI ≥ 40 at any point during their CPRD registration, which could span 10 years or more. It is therefore possible that the BMI result recorded closest to the index date was lower than 40 and, as you note, the baseline results show this to be the case. Where the closest recorded BMI to baseline is <35 , this could be for a number of reasons: 1) the measure is accurate – this could even apply to surgery patients, particularly those receiving surgery outside National Health Service provision; 2) the measure could be from an earlier time point, and may not accurately reflect their BMI at index; or 3) the record could be an error. We have conducted a sensitivity analysis excluding those in this category and the result is comparable to the main analysis: crude rate ratio 0.63, 95% CI 0.40; 0.99; adjusted rate ratio: 0.39, 95%CI 0.23; 0.65. We have added this result to the manuscript in the S4 appendix “Sensitivity analyses for the association of bariatric surgery with acute kidney injury”.

Baseline BMI can only be ascertained if recorded by the GP and hence is missing for a small proportion of patients.

2 - The authors do point out the limitations of AKI ascertainment with ICD codes, and how it is dependent on testing. Baseline AKI differed between the groups and is also dependent on the length of follow-up prior to the baseline period. Did this differ between groups? One could consider using time of follow-up before the baseline period as a matching variable or a metric capturing exposure to the health system (e.g. # of outpatient visits, hospitalizations)

Reply: The reviewer raises a good point about potential differential ascertainment of AKI at baseline where we observed 1.1% of exposed patients with a history of AKI compared to 0.4% of unexposed. The median follow-up time prior to baseline was very similar in the two groups; (8.8 years [IQR: 8.1 years] for exposed patients and 9.3 years [IQR: 8.0 years] for unexposed). Thus, the difference in history of AKI at baseline between the exposed and unexposed is unlikely due to differential follow-up prior to the study. This has now been added to the results section of the manuscript (page 11, lines 231-233):

“The median follow-up prior to baseline was similar between the groups: 8.8 years (IQR: 8.1 years) for patients with bariatric surgery and 9.3 years (IQR: 8.0 years) for people without surgery.”

3 - As the authors point out a major limitation was the lack of baseline CKD status in many patients and also the fact that 40% of patients get bariatric surgery in the private setting. Can the authors comment on the chances these individuals are receiving follow-up care in the private setting as well.

Reply: In England, the vast majority of people access healthcare through the NHS, the main reason for seeking private care is to avoid waiting for routine surgical intervention. We assume that people with and without surgery are equally likely to visit an NHS hospital, especially for acute and high-risk events associated with AKI (e.g. severe chest infection, sepsis), as acute emergency care in the NHS is accessible. Moreover, private providers often refer complex patients (e.g. those with AKI) back to NHS hospitals where there is greater experience and better infrastructure for dealing with such cases. Is there any surrogate for socioeconomic status in the dataset that can be assessed?

Reply: Unfortunately, surrogate measures for socioeconomic status are not available in the dataset for this study.

For limitations, misclassification of diagnostic codes could be differential if there is a pre-conceived notion that bariatric surgery patients are at increased risk of AKI.

Reply: We agree with this hypothesis, however, in our experience, clinical concern regarding AKI is related to immediate post-operative complications, rather than in long-term follow-up. Initially, patients with bariatric surgery may have been under more scrutiny to detect potential AKI events than patients without surgery. Thus our current relative risk estimate in the immediate post-surgical period would be an overestimate. We have amended the discussion accordingly (page 16, lines 299-302):

“However, it is also conceivable during the immediate post-operative period those undergoing bariatric surgery might have been under more scrutiny to detect potential AKI events than people without surgery. In this case our current relative risk estimate for the immediate postoperative period would be an overestimate.”

Lastly, why is baseline eGFR missing for patients undergoing bariatric surgery? Is it possible they may have labs drawn elsewhere outside of the NHS? If so, then might they get follow-up labs also outside of the NHS?

Reply: As discussed above: patients undergoing bariatric surgery are likely to have had preoperative assessment of their kidney function in secondary care but the GP would have only been informed by the hospital if the result had been abnormal. Thus an absence of eGFR means that these patients are very unlikely to have an eGFR < 60 ml/min/1.73m².

VERSION 2 – REVIEW

REVIEWER	Prof. Dr. Helmut Schiffl Department of Internal Medicine Medizinische Klinik IV University Hospital Munich
REVIEW RETURNED	21-Dec-2017

GENERAL COMMENTS	I regret deeply to have to reject the manuscript, as the authors have not improved their manuscript and did not respond adequately to the concerns raised
---

REVIEWER	Alex Chang Geisinger USA
REVIEW RETURNED	23-Dec-2017

GENERAL COMMENTS	The author have addressed my concerns
---------------------------------------

VERSION 2 – AUTHOR RESPONSE

We thank the editor for the suggestions and have amended the manuscript accordingly:

- the "Strengths and limitations" section now reflects limitations of the ICD-10 codes (lines 61/62)
- we addressed the limitations of ICD-10 codes in the discussion section (lines 298-309)
- we have toned down the conclusions in the abstract section (line 48)
